# Infinite Factorial Dynamical Model

**Isabel Valera**[*]
Max Planck Institute for
Software Systems
ivalera@mpi-sws.org

**Francisco J. R. Ruiz**[*]
Department of Computer Science
Columbia University
f.ruiz@columbia.edu

**Lennart Svensson**
Department of Signals and Systems
Chalmers University of Technology
lennart.svensson@chalmers.se

**Fernando Perez-Cruz**
Universidad Carlos III de Madrid, and
Bell Labs, Alcatel-Lucent
fernandop@ieee.org

## Abstract

We propose the infinite factorial dynamic model (iFDM), a general Bayesian non-parametric model for source separation. Our model builds on the Markov Indian buffet process to consider a potentially unbounded number of hidden Markov chains (sources) that evolve independently according to some dynamics, in which the state space can be either discrete or continuous. For posterior inference, we develop an algorithm based on particle Gibbs with ancestor sampling that can be efficiently applied to a wide range of source separation problems. We evaluate the performance of our iFDM on four well-known applications: multitarget tracking, cocktail party, power disaggregation, and multiuser detection. Our experimental results show that our approach for source separation does not only outperform previous approaches, but it can also handle problems that were computationally intractable for existing approaches.

## 1 Introduction

The central idea behind Bayesian nonparametrics (BNPs) is the replacement of classical finite-dimensional prior distributions with general stochastic processes, allowing for an open-ended number of degrees of freedom in a model [8]. They constitute an approach to model selection and adaptation in which the model complexity is allowed to grow with data size [17]. In the literature, BNP priors have been applied for time series modeling. For example, the infinite hidden Markov model [2, 20] considers a potentially infinite cardinality of the state space; and the BNP construction of switching linear dynamical systems (LDS) [4] considers an unbounded number of dynamical systems with transitions among them occurring at any time during the observation period.

In the context of signal processing, the source separation problem has captured the attention of the research community for decades due to its wide range of applications [12, 23, 7, 24]. The BNP literature for source separation includes [10], in which the authors introduce the nonparametric counterpart of independent component analysis (ICA), referred as infinite ICA (iICA); and [23], where the authors present the Markov Indian buffet process (mIBP), which places a prior over an infinite number of parallel Markov chains and is used to build the infinite factorial hidden Markov model (iFHMM) and the ICA iFHMM. These approaches can effectively adapt the number of hidden sources to fit the available data. However, they suffer from several limitations: i) the iFHMM is restricted to binary on/off hidden states, which may lead to hidden chains that do not match the actual hidden causes, and it is not able to deal with continuous-valued states, and ii) both the iICA and the ICA iFHMM make independence assumptions between consecutive values of active hidden states, which significantly restricts their ability to capture the underlying dynamical models. As a result, we find that existing approaches are not applicable to many well-known source separation

---

[*] Both authors contributed equally.

problems, such as multitarget tracking [12], in which each target can be modeled as a Markov chain with continuous-valued states describing the target trajectory; or multiuser detection [24], in which the high cardinality of the hidden states makes this problem computationally intractable for the non-binary extension of the iFHMM. Hence, there is a lack of both a general BNP model for source separation, and an efficient inference algorithm to address these limitations.

In this paper, we provide a general BNP framework for source separation that can handle a wide range of dynamics and likelihood models. We assume a potentially infinite number of sources that are modeled as Markov chains that evolve according to some dynamical system model. We assume that only the active sources contribute to the observations, and the states of the Markov chains are not restricted to be discrete but they can also be continuous-valued. Moreover, we let the observations depend on both the current state of the hidden sequences, and on some previous states. This system memory is needed when dealing with applications in which the individual source signals propagate through the air and may thus suffer from some phenomena, such as reverberation, echo, or multipath propagation. Our approach results in a general and flexible dynamic model that we refer to as infinite factorial dynamical model (iFDM), and that can be particularized to recover other models previously proposed in the literature, e.g., the binary iFHMM.

As for most BNP models, one of the main challenges of our iFDM is posterior inference. In discrete time series models, including the iFHMM, an approximate inference algorithm based on forward-filtering backward-sampling (FFBS) sweeps is typically used [23, 5]. However, the exact FFBS algorithm has exponential computational complexity with respect to the memory length. The FFBS algorithm also becomes computationally intractable when dealing with on/off hidden states that are continuous-valued when active. In order to overcome these limitations, we develop a suitable inference algorithm for our iFDM by building a Markov chain Monte Carlo (MCMC) kernel using particle Gibbs with ancestor sampling (PGAS) [13]. This algorithm presents quadratic complexity with respect to the memory length and can easily handle a broad range of dynamical models.

The versatility and efficiency of our approach is shown through a comprehensive experimental validation in which we tackle four well-known source separation problems: multitarget tracking [12], cocktail party [23], power disaggregation [7], and multiuser detection [24].[1] Our results show that our iFDM provides meaningful estimations of the number of sources and their corresponding individual signal traces even in applications that previous approaches cannot handle. It also outperforms, in terms of accuracy, the iFHMM (extended to account for the actual state space cardinality) combined with FFBS-based inference in the cocktail party and power disaggregation problems.

## 2   Infinite Factorial Dynamical Model

In this section, we detail our proposed iFDM. We assume that there is a potentially infinite number of sources contributing to the observed sequence $\{\mathbf{y}_t\}_{t=1}^T$, and each source is modeled by an underlying dynamic system model in which the state of the $m$-th source at time $t$, denoted by $x_{tm} \in \mathcal{X}$, evolves over time as a first-order Markov chain. Here, the state space $\mathcal{X}$ can be either discrete or continuous. In addition, we introduce the auxiliary binary variables $s_{tm} \in \{0, 1\}$ to indicate whether the $m$-th source is active at time $t$, such that the observations only depend on the active sources. We assume that the variables $s_{tm}$ follow a first-order Markov chain and let the states $x_{tm}$ evolve according to $p(x_{tm}|s_{tm}, x_{(t-1)m}, s_{(t-1)m})$, i.e., the dynamic system model may depend on whether the source is active or inactive. We assume dummy states $s_{tm} = 0$ for $t \leq 0$. As an example, in the cocktail party problem, $\mathbf{y}_t$ denotes a sample of the recorded audio signal, which depends on the individual voice signals of the active speakers. The latent states $x_{tm}$ in this example are real-valued and the transition model $p(x_{tm}|s_{tm} = 1, x_{(t-1)m}, s_{(t-1)m})$ describes the dynamics of the voice signal.

In many real applications, the individual signals propagate though the air until they are mixed and gathered by the receiver. In such propagation, different phenomena (e.g., refraction or reflexion of the signal in the walls) may occur, leading to multipath propagation of the signals and, therefore, to different delayed copies of the individual signals at the receiver. In order to account for this "memory" effect, we consider that the state of the $m$-th source at time $t$, $x_{tm}$, influences not only the observation $\mathbf{y}_t$, but also the future $L - 1$ observations, $\mathbf{y}_{t+1}, \ldots, \mathbf{y}_{t+L-1}$. Therefore, the likelihood of $\mathbf{y}_t$ depends on the last $L$ states of all the Markov chains, yielding

$$p(\mathbf{y}_t|\mathbf{X}, \mathbf{S}) = p(\mathbf{y}_t|\{x_{tm}, s_{tm}, x_{(t-1)m}, s_{(t-1)m}, \ldots, x_{(t-L+1)m}, s_{(t-L+1)m}\}_{m=1}^M), \quad (1)$$

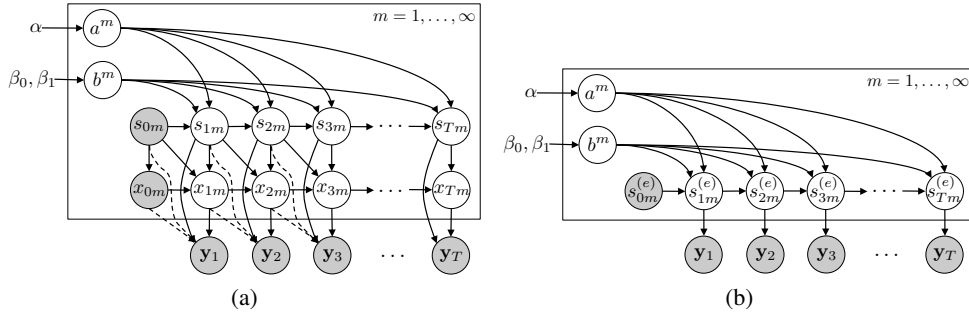

(a)                      (b)

Figure 1: (a) Graphical representation of the iFDM with memory length $L = 2$. The dashed lines represent the memory. (b) Equivalent representation using extended states.

where $\mathbf{X}$ and $\mathbf{S}$ are $T \times M$ matrices containing all the states $x_{tm}$ and $s_{tm}$, respectively. We remark that the likelihood of $\mathbf{y}_t$ cannot depend on any hidden state $x_{\tau m}$ if $s_{\tau m} = 0$.

In order to be able to deal with an infinite number of sources, we place a BNP prior over the binary matrix $\mathbf{S}$ that contains all variables $s_{tm}$. In particular, we assume that $\mathbf{S} \sim \text{mIBP}(\alpha, \beta_0, \beta_1)$, i.e., $\mathbf{S}$ is distributed as a mIBP [23] with parameters $\alpha$, $\beta_0$ and $\beta_1$. The mIBP places a prior distribution over binary matrices with a finite number of rows $T$ and an infinite number of columns $M$, in which each row represents a time instant, and each column represents a Markov chain. The mIBP ensures that, for any finite value of $T$, only a finite number of columns $M_+$ in $\mathbf{S}$ are active almost surely, whereas the rest of them remain in the all-zero state and do not influence the observations. We make use of the stick-breaking construction of the mIBP, which is particularly useful to develop many practical inference algorithms [19, 23]. Under the stick-breaking construction, two hidden variables for each Markov chain are introduced, representing the transition probabilities between the active and inactive states. In particular, we define $a^m = p(s_{tm} = 1|s_{(t-1)m} = 0)$ as the transition probability from inactive to active, and $b^m = p(s_{tm} = 1|s_{(t-1)m} = 1)$ as the self-transition probability of the active state of the $m$-th chain. In the stick-breaking representation, the columns of $\mathbf{S}$ are ordered according to their values of $a^m$, such that $a^1 > a^2 > a^3 > \ldots$, and the probability distribution over variables $a^m$ is given by $a^1 \sim \text{Beta}(\alpha, 1)$, and $p(a^m|a^{m-1}) \propto (a^m)^{\alpha-1}\mathbb{I}(0 \leq a^m \leq a^{m-1})$, being $\mathbb{I}(\cdot)$ the indicator function [19]. Finally, we place a beta distribution over the transition probabilities $b^m$ of the form $b^m \sim \text{Beta}(\beta_0, \beta_1)$.

The resulting iFDM model, particularized for $L = 2$, is shown in Figure 1a. Note that this model can be equivalently represented as shown in Figure 1b, using the extended states $s_{tm}^{(e)}$, with

$$s_{tm}^{(e)} = \left[ \begin{array}{cccccc} x_{tm}, & s_{tm}, & x_{(t-1)m}, & s_{(t-1)m}, & \ldots, & x_{(t-L+1)m}, & s_{(t-L+1)m} \end{array} \right]. \quad (2)$$

This extended representation allows for an FFBS-based inference algorithm. However, the exponential complexity of the FFBS with the memory parameter $L$ and with continuous-valued hidden states $x_{tm}$ makes the algorithm intractable in many real scenarios. Hence, we maintain the representation in Figure 1a because it allows us to derive an efficient inference algorithm.

The proposed iFDM in Figure 1a can be particularized to resemble some other models that have been proposed in the literature. In particular, we recover: i) the iFHMM in [23] by choosing the state space $\mathcal{X} = \{0, 1\}$, $x_{tm} = s_{tm}$ and $L = 1$, ii) the ICA iFHMM in [23] if we set $\mathcal{X} = \mathbb{R}$, $L = 1$ and assume that $p(x_{tm}|s_{tm} = 1, x_{(t-1)m}, s_{(t-1)m}) = p(x_{tm}|s_{tm} = 1)$ is a Gaussian distribution, and iii) a BNP counterpart of the LDS [9] with on/off states by assuming $L = 1$ and $\mathcal{X} = \mathbb{R}$, and letting the variables $x_{tm}$ be Gaussian distributed with linear relationships among them.

## 3 Inference Algorithm

We develop an inference algorithm for the proposed iFDM that can handle different dynamic and likelihood models. Our approach relies on a blocked Gibbs sampling algorithm that alternates between sampling the number of considered chains and the global variables conditioned on the current value of matrices $\mathbf{S}$ and $\mathbf{X}$, and sampling matrices $\mathbf{S}$ and $\mathbf{X}$ conditioned on the current value of the remaining variables. In particular, the algorithm proceeds iteratively as follows:

- **Step 1:** Add $M_{\text{new}}$ new inactive chains using an auxiliary slice variable and a slice sampling method. In this step, the number of considered chains is increased from its initial value $M_+$ to $M^{\ddagger} = M_+ + M_{\text{new}}$ ($M_+$ is not updated because $s_{tm} = 0$ for all $t$ for the new chains).

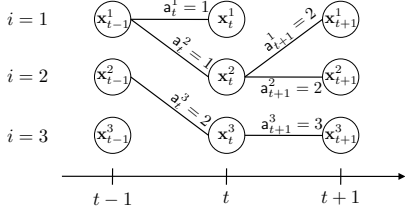

$i = 1$ ... $a_t^1 = 1$ ... $a_{t+1}^1 = 2$

$i = 2$ ... $a_t^2 = 1$ ... $a_{t+1}^2 = 2$

$i = 3$ ... $a_t^3 = 2$ ... $a_{t+1}^3 = 3$

$t-1$      $t$      $t+1$

(a) Example of the connection of particles in PGAS. We represent $P = 3$ particles $\mathbf{x}_\tau^i$ for $\tau = \{t-1, t, t+1\}$. The index $a_\tau^i$ denotes the ancestor particle of $\mathbf{x}_\tau^i$. It can be seen that, e.g., the trajectories $\mathbf{x}_{1:t+1}^1$ and $\mathbf{x}_{1:t+1}^2$ only differ at time instant $t+1$.

---

**Input** : Reference particle $\mathbf{x}_t'$ for $t = 1, \ldots, T$, and global variables.
**Output**: Sample $\mathbf{x}_{1:T}^{\text{out}}$ from the PGAS Markov kernel

1   Draw $\mathbf{x}_1^i \sim r_1(\mathbf{x}_1)$ for $i = 1, \ldots, P-1$ (Eq. 4)
2   Set $\mathbf{x}_1^P = \mathbf{x}_1'$
3   Compute the weights $w_1^i = W_1(\mathbf{x}_1^i)$ for $i = 1, \ldots, P$ (Eq. 5)
4   **for** $t = 2, \ldots, T$ **do**
       // Resampling and ancestor sampling
5      Draw $a_t^i \sim \text{Categorical}(w_{t-1}^1, \ldots, w_{t-1}^P)$ for $i = 1, \ldots, P-1$
6      Compute $\widetilde{w}_{t-1|T}^i$ for $i = 1, \ldots, P$ (Eq. 6)
7      Draw $a_t^P \sim \text{Categorical}(\widetilde{w}_{t-1|T}^1, \ldots, \widetilde{w}_{t-1|T}^P)$
       // Particle propagation
8      Draw $\mathbf{x}_t^i \sim r_t(\mathbf{x}_t | \mathbf{x}_{1:t-1}^{a_t^i})$ for $i = 1, \ldots, P-1$ (Eq. 4)
9      Set $\mathbf{x}_t^P = \mathbf{x}_t'$
10     Set $\mathbf{x}_{1:t}^i = (\mathbf{x}_{1:t-1}^{a_t^i}, \mathbf{x}_t^i)$ for $i = 1, \ldots, P$ (Eq. 3)
       // Weighting
11     Compute the weights $w_t^i = W_t(\mathbf{x}_{1:t}^i)$ for $i = 1, \ldots, P$ (Eq. 5)
12   Draw $k \sim \text{Categorical}(w_T^1, \ldots, w_T^P)$
13   **return** $\mathbf{x}_{1:T}^{\text{out}} = \mathbf{x}_{1:T}^k$

---

(b) PGAS algorithm.

Figure 2: Particle Gibbs with ancestor sampling.

- **Step 2:** Jointly sample the states $x_{tm}$ and $s_{tm}$ of all the considered chains. Compact the representation by removing those chains that remain inactive in the entire observation period, consequently updating $M_+$.
- **Step 3:** Sample the global variables in the model, which include the transition probabilities and the emission parameters, from their posterior distribution.

In **Step 1**, we follow the slice sampling scheme for inference in BNP models based on the Indian buffet process (IBP) [19, 23], which effectively transforms the model into a finite factorial model with $M^\ddagger = M_+ + M_{\text{new}}$ parallel chains. **Step 2** consists in sampling the elements of the matrices $\mathbf{S}$ and $\mathbf{X}$ given the current value of the global variables. Here, we propose to use PGAS, an algorithm recently developed for inference in state-space models and non-Markovian latent variable models [13]. Each iteration of this algorithm presents quadratic complexity with respect to the memory length $L$, avoiding the exponential complexity of the standard FFBS algorithm when applied over the equivalent model with extended states in Figure 1b. Details on the PGAS approach are given in Section 3.1. After running PGAS, we remove those chains that remain inactive in the whole observation period. In **Step 3**, we sample the transition probabilities $a^m$ and $b^m$, as well as other model-dependent variables such as the observation variables needed to evaluate the likelihood $p(\mathbf{y}_t | \mathbf{X}, \mathbf{S})$. Further details on the inference algorithm can be found in the Supplementary Material.

### 3.1 Particle Gibbs with ancestor sampling

PGAS [13] is a method within the framework of particle MCMC [1] that combines the main ideas, as well as the strengths, of sequential Monte Carlo and MCMC techniques. In contrast to other particle Gibbs with backward simulation methods [25, 14], this algorithm can also be conveniently applied to non-Markovian latent variable models, i.e., models that are not expressed on a state-space form. The PGAS algorithm is an MCMC kernel, and thus generates a new sample of the hidden state matrices $(\mathbf{X}, \mathbf{S})$ given an initial sample $(\mathbf{X}', \mathbf{S}')$, which is the output of the previous iteration of the PGAS (extended to account for the $M_{\text{new}}$ new inactive chains). The machinery inside the PGAS algorithm resembles an ordinary particle filter, with two main differences: one of the particles is deterministically set to the reference input sample, and the ancestor of each particle is randomly chosen and stored during the algorithm execution. We briefly describe the PGAS approach below, but we refer to [13] for a rigorous analysis of the algorithm properties.

In the proposed PGAS, we assume a set of $P$ particles for each time instant, each representing the states $\{x_{tm}, s_{tm}\}_{m=1}^{M^\ddagger}$. We denote by the vector $\mathbf{x}_t^i$ the state of the $i$-th particle at time $t$. We also introduce the ancestor indexes $a_t^i \in \{1, \ldots, P\}$ in order to denote the particle that precedes the $i$-th particle at time $t$. That is, $a_t^i$ corresponds to the index of the ancestor particle of $\mathbf{x}_t^i$. Let also $\mathbf{x}_{1:t}^i$ be the ancestral path of particle $\mathbf{x}_t^i$, i.e., the particle trajectory that is recursively defined as $\mathbf{x}_{1:t}^i = (\mathbf{x}_{1:t-1}^{a_t^i}, \mathbf{x}_t^i)$. Figure 2a shows an example to clarify the notation.

The algorithm is summarized in Figure 2b. For each time instant $t$, we first generate the ancestor indexes for the first $P-1$ particles according to the importance weights $w_{t-1}^i$. Given these ancestors, the particles are then propagated across time according to a distribution $r_t(\mathbf{x}_t|\mathbf{x}_{1:t-1})$. For simplicity, and dropping the global variables from the notation for conciseness, we assume that

$$r_t(\mathbf{x}_t|\mathbf{x}_{1:t-1}^{\mathsf{a}_t}) = p(\mathbf{x}_t|\mathbf{x}_{t-1}^{\mathsf{a}_t}) = \prod_{m=1}^{M^{\ddagger}} p(x_{tm}|s_{tm}, x_{(t-1)m}^{\mathsf{a}_t}, s_{(t-1)m}^{\mathsf{a}_t}) p(s_{tm}|s_{(t-1)m}^{\mathsf{a}_t}), \qquad (3)$$

i.e., particles are propagated as in Figure 1a using a simple bootstrap proposal kernel, $p(x_{tm}, s_{tm}|s_{(t-1)m}, x_{(t-1)m})$. The $P$-th particle is instead deterministically set to the reference particle, $\mathbf{x}_t^P = \mathbf{x}_t'$, whereas the ancestor indexes $\mathsf{a}_t^P$ are sampled according to some weights $\widetilde{w}_{t-1|T}^i$. Indeed, this is a crucial step that vastly improves the mixing properties of the MCMC kernel.

We now focus on the computation on the importance weights $w_t^i$ and the ancestor weights $\widetilde{w}_{t-1|T}^i$. For the former, the particles are weighted according to $w_t^i = W_t(\mathbf{x}_{1:t}^i)$, where

$$W_t(\mathbf{x}_{1:t}) = \frac{p(\mathbf{x}_{1:t}|\mathbf{y}_{1:t})}{p(\mathbf{x}_{1:t-1}|\mathbf{y}_{1:t-1}) r_t(\mathbf{x}_t|\mathbf{x}_{1:t-1})} \propto p(\mathbf{y}_t|\mathbf{x}_{t-L+1:t}), \qquad (4)$$

being $\mathbf{y}_{\tau_1:\tau_2}$ the set of observations $\{\mathbf{y}_t\}_{t=\tau_1}^{\tau_2}$. Eq. 4 implies that, in order to obtain the importance weights, it suffices to evaluate the likelihood at time $t$. The weights $\widetilde{w}_{t-1|T}^i$ are given by

$$\widetilde{w}_{t-1|T}^i = w_{t-1}^i \frac{p(\mathbf{x}_{1:t-1}^i, \mathbf{x}_{t:T}'|\mathbf{y}_{1:T})}{p(\mathbf{x}_{1:t-1}^i|\mathbf{y}_{1:t-1})} \propto w_{t-1}^i p(\mathbf{x}_t'|\mathbf{x}_{t-1}^i) \prod_{\tau=t}^{t+L-2} p(\mathbf{y}_\tau|\mathbf{x}_{1:t-1}^i, \mathbf{x}_{t:T}'). \qquad (5)$$

Note that, for memoryless models (i.e., $L=1$), Eq. 5 can be simplified, since the product in the last term is not present and, therefore, $\widetilde{w}_{t-1|T}^i \propto w_{t-1}^i p(\mathbf{x}_t'|\mathbf{x}_{t-1}^i)$. For $L > 1$, the computation of the weights $\widetilde{w}_{t-1|T}^i$ in (5) for $i = 1, \ldots, P$ has computational time complexity scaling as $\mathcal{O}(PM^{\ddagger}L^2)$. Since this computation needs to be performed for each time instant (and this is the most expensive calculation), the resulting algorithm complexity scales as $\mathcal{O}(PTM^{\ddagger}L^2)$.

## 4 Experiments

We now evaluate the proposed model and inference algorithm on four different applications, which are detailed below and summarized in Table 1. For the PGAS kernel, we use $P = 3,000$ particles in all our experiments. Additional details on the experiments are given in the Supplementary Material.

**Multitarget Tracking.** In the multitarget tracking problem, we aim at locating the position of several moving targets based on noisy observations. Under a general setup, a varying number of indistinguishable targets are moving around in a region, appearing at random in space and time. Multitarget tracking plays an important role in many areas of engineering such as surveillance, computer vision and signal processing [18, 16, 21, 6, 12]. Here, we focus on a simple synthetic example to show that our proposed iFDM can handle time-dependent continuous-valued hidden states. We place three moving targets within a region of $800 \times 800$ metres, where 25 sensors are located on a square grid. The state $\boldsymbol{x}_{tm} = [x_{tm}^{(1)}, x_{tm}^{(2)}, v_{tm}^{(1)}, v_{tm}^{(2)}]^{\top}$ of each target consists of its position and velocity in a two dimensional plane, and we assume a linear Gaussian dynamic model such that, while active, $\boldsymbol{x}_{tm}$ evolves according to

$$\boldsymbol{x}_{tm} = \mathbf{G}_x \boldsymbol{x}_{(t-1)m} + \mathbf{G}_u \mathbf{u}_t \qquad (6)$$

where $\mathbf{G}_x = [1\,0\,T_s\,0; 0\,1\,0\,T_s; 0\,0\,1\,0; 0\,0\,0\,1]$, $\mathbf{G}_u = [\frac{T_s^2}{2}\,0; 0\,\frac{T_s^2}{2}; T_s\,0; 0\,T_s]$, $T_s = 0.5$ is the sampling period, and $\mathbf{u}_t \sim \mathcal{N}(\mathbf{0}, \mathbf{I})$ is a vector that models the acceleration noise. For each considered target, we sample the initial position uniformly in the sensor network space, and assume that the initial velocity is Gaussian distributed with zero mean and covariance $0.01\mathbf{I}$. Following [21, 12], we generate ($T = 300$) observations based on the received signal strength (RSS), where the measurement of sensor $j$ at time $t$ is given by $y_{tj} = \sum_{m:s_{tm}=1} P_0 \cdot \left(\frac{d_0}{d_{mjt}}\right)^{\gamma} + n_{tj}$. Here, $n_{tj} \sim \mathcal{N}(0, 2)$ is the noise term, $d_{mjt}$ is the distance between target $m$ and sensor $j$ at time $t$, $P_0 = 10$ is the transmitted power, and $d_0 = 100$ metres and $\gamma = 2$ are, respectively, the reference distance and the path loss exponent, which account for the radio propagation model. In our inference algorithm, we sample the noise variance by placing an InvGamma(1,1) distribution as its prior. Here, we compare

| Application | Model | $\mathcal{X}$ | $p(x_{tm}|s_{tm} = 1, x_{(t-1)m}, s_{(t-1)m} = 1)$ | $L$ |
|---|---|---|---|---|
| Multitarget Tracking | Infinite factorial LDS | $\mathbb{R}^4$ | $\mathcal{N}(\boldsymbol{x}_{tm}|\mathbf{G}_x\boldsymbol{x}_{(t-1)m}, \mathbf{G}_u\mathbf{G}_u^\top)$ | 1 |
| Cocktail Party | ICA iFHMM | $\mathbb{R}$ | $\mathcal{N}(x_{tm}|0, \sigma_x^2)$ | 1 |
| Power Dissagregation | Non-binary iFHMM | $\{0, 1, \ldots, Q - 1\}$ | $a_{jk}^m = p(x_{tm} = k|x_{(t-1)m} = j)$ | 1 |
| Multiuser Detection | – | $\mathcal{A}\bigcup\{0\}$ | $\mathcal{U}(\mathcal{A})$ | $\in \mathbb{N}$ |

Table 1: Applications of the iFDM.

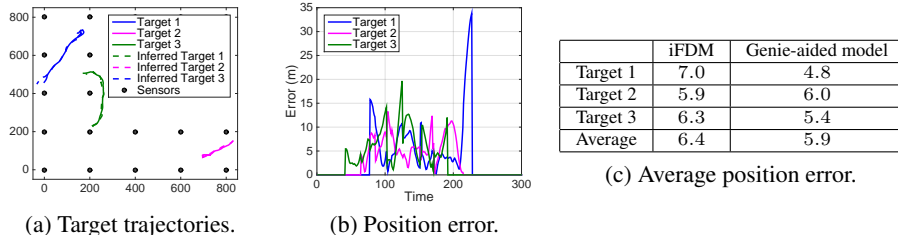

(a) Target trajectories.

(b) Position error.

|  | iFDM | Genie-aided model |
|---|---|---|
| Target 1 | 7.0 | 4.8 |
| Target 2 | 5.9 | 6.0 |
| Target 3 | 6.3 | 5.4 |
| Average | 6.4 | 5.9 |

(c) Average position error.

Figure 3: Results for the multitarget tracking problem.

the performance of the iFDM with a 'genie-aided' finite factorial model with perfect knowledge of the number of targets and noise variance.

In Figures 3a and 3b, we show the true and inferred trajectories of the targets, and the temporal evolution of the position error of the iFDM. Additionally, Figure 3c shows the average position error (in absolute value) for our iFDM and the genie-aided method. In these figures, we observe that the proposed model and algorithm is able to detect the three targets and their trajectories, providing similar performance to the genie-aided method. In particular, both approaches provide average position errors of around 6 metres, which is thrice the noise variance.

**Cocktail Party.** We now address a blind speech separation task, also known as the cocktail party problem. Given the recorded audio signals from a set of microphones, the goal is to separate out the individual speech signals of multiple people who are speaking simultaneously. Speakers may start speaking or become silent at any time. Similarly to [23], we collect data from several speakers from the PASCAL 'CHiME' Speech Separation and Recognition Challenge website.[2] The voice signal for each speaker consists of 4 sentences, which we append with random pauses in between each sentence. We artificially mix the data 10 times (corresponding to 10 microphones) with mixing weights sampled from Uniform$(0, 1)$, such that each microphone receives a linear combination of all the considered signals, corrupted by Gaussian noise with standard deviation $0.3$. We consider two scenarios, with 5 and 15 speakers, and subsample the data so that we learn from $T = 1,354$ and $T = 1,087$ datapoints, respectively. Following [23], our model assumes $p(x_{tm}|s_{tm} = 1, x_{(t-1)m}, s_{(t-1)m}) = \mathcal{N}(x_{tm}|0, 2)$, and $x_{tm} = 0$ whenever $s_{tm} = 0$. We also model $\mathbf{y}_t$ as a linear combination of all the voice signals under Gaussian noise, i.e., $\mathbf{y}_t = \sum_{m=1}^{M_+} \mathbf{w}_m x_{tm} + \mathbf{n}_t$, where $\mathbf{n}_t \sim \mathcal{N}(\mathbf{0}, \sigma_y^2\mathbf{I})$ is the noise term, $\mathbf{w}_m \sim \mathcal{N}(\mathbf{0}, \mathbf{I})$ is the 10-dimensional weighting vector associated to the $m$-th speaker, and $\sigma_y^2 \sim \text{InvGamma}(1, 1)$. We compare our iFDM with the ICA iFHMM in [23] using FFBS sweeps for inference, with (i) $p(x_{tm}|s_{tm} = 1) = \mathcal{N}(x_{tm}|0, 2)$ (denoted as FFBS-G), and (ii) $p(x_{tm}|s_{tm} = 1) = \text{Laplace}(x_{tm}|0, 2)$ (denoted as FFBS-L).

For the scenario with 5 speakers, we show the true and the inferred (after iteration $10,000$) number of speakers in Figures 4a, 4b, 4c and 4d, along with their activities during the observation period. In order to quantitatively evaluate the performance of the different algorithms, we show in Figure 4e (top) the activity detection error rate (ADER), which is computed as the probability of detecting activity (inactivity) of a speaker while that speaker is actually inactive (active). As the algorithms are unsupervised, we sort the estimated chains so that the ADER is minimized. If the inferred number of speakers $M_+$ is smaller (larger) than the true number of speakers, we consider some extra inferred inactive chains (additional speakers). The PGAS-based approach outperforms the two FFBS-based methods because it can jointly sample the states of all chains (speakers) for each time instant, whereas the FFBS requires sampling each chain conditioned on the current states of the other chains, leading to poor mixing, as discussed in [22]. As a consequence, the FFBS tends to overestimate the number of speakers, as shown in Figure 4e (bottom).

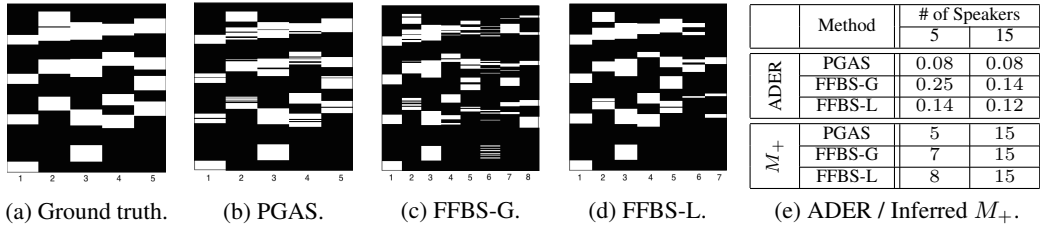

| | Method | # of Speakers | |
|---|---|---|---|
| | | 5 | 15 |
| ADER | PGAS | 0.08 | 0.08 |
| | FFBS-G | 0.25 | 0.14 |
| | FFBS-L | 0.14 | 0.12 |
| $M_+$ | PGAS | 5 | 15 |
| | FFBS-G | 7 | 15 |
| | FFBS-L | 8 | 15 |

(a) Ground truth.  (b) PGAS.  (c) FFBS-G.  (d) FFBS-L.  (e) ADER / Inferred $M_+$.

Figure 4: Results for the cocktail party problem.

| Algorithm | H. 1 | H. 2 | H. 3 | H. 4 | H. 5 |
|---|---|---|---|---|---|
| PGAS | 0.68 | 0.79 | 0.60 | 0.58 | 0.55 |
| FFBS | 0.59 | 0.78 | 0.56 | 0.53 | 0.43 |

(a) REDD ('H' stands for 'House').

| Algorithm | Day 1 | Day 2 |
|---|---|---|
| PGAS | 0.76 | 0.82 |
| FFBS | 0.67 | 0.72 |

(b) AMP.

Table 2: Accuracy for the power disaggregation problem.

**Power Disaggregation.** Given the aggregate whole-home power consumption signal, the power disaggregation problem consists in estimating both the number of active devices in the house and the power draw of each individual device [11, 7]. We validate the performance of the iFDM on two different real databases: the Reference Energy Disaggregation Data Set (REDD) [11], and the Almanac of Minutely Power Dataset (AMP) [15]. For the AMP database, we consider two 24-hour segments and 8 devices. For the REDD database, we consider a 24-hour segment across 5 houses and 6 devices. Our model assumes that each device can take $Q = 4$ different states (one inactive state and three active states with different power consumption), i.e., $x_{tm} \in \{0, 1, \ldots, Q-1\}$, with $x_{tm} = 0$ if $s_{tm} = 0$. We place a symmetric Dirichlet prior over the transition probability vectors of the form $\mathbf{a}_j^m \sim \text{Dirichlet}(1)$, where each element $a_{jk}^m = p(x_{tm} = k | s_{tm} = 1, x_{(t-1)m} = j, s_{(t-1)m})$. When $x_{tm} = 0$, the power consumption of device $m$ at time $t$ is zero ($P_0^m = 0$), and when $x_{tm} \in \{1, \ldots, Q-1\}$ its average power consumption is given by $P_{x_{tm}}^m$. Thus, the total power consumption is given by $y_t = \sum_{m=1}^{M_+} P_{x_{tm}}^m + n_t$, where $n_t \sim \mathcal{N}(0, 0.5)$ represents the additive Gaussian noise. For $q \in \{1, \ldots, Q-1\}$, we assume a prior power consumption $P_q^m \sim \mathcal{N}(15, 10)$. In this case, the proposed model for the iFDM resembles a non-binary iFHMM and, therefore, we can also apply the FFBS algorithm to infer the power consumption draws of each device.

In order to evaluate the performance of the different algorithms, we compute the mean accuracy of the estimated consumption of each device (higher is better), i.e., $\text{acc} = 1 - \frac{\sum_{t=1}^{T} \sum_{m=1}^{M} |x_t^{(m)} - \hat{x}_t^{(m)}|}{2 \sum_{t=1}^{T} \sum_{m=1}^{M} x_t^{(m)}}$, where $x_t^{(m)}$ and $\hat{x}_t^{(m)} = P_{x_{tm}}^m$ are, respectively, the true and the estimated power consumption by device $m$ at time $t$. In order to compute the accuracy, we assign each estimated chain to a device so that the accuracy is maximized. If the inferred number of devices $M_+$ is smaller than the true number of devices, we use $\hat{x}_t^{(m)} = 0$ for the undetected devices. If $M_+$ is larger than the true number of devices, we group all the extra chains as an "unknown" device and use $x_t^{(\text{unk})} = 0$. In Table 2 we show the results provided by both algorithms. The PGAS approach outperforms the FFBS algorithm in the five houses of the REDD database and the two selected days of the AMP database. This occurs because the PGAS can simultaneously sample the hidden states of all devices for each time instant, whereas the FFBS requires conditioning on the current states of all but one device.

**Multiuser Detection.** We now consider a digital communication system in which users are allowed to enter or leave the system at any time, and several receivers cooperate to estimate the number of users, the (digital) symbols they transmit, and the propagation channels they face. Multipath propagation affects the radio signal, thus causing inter-symbol interference. To capture this phenomenon in our model, we use $L \geq 1$ in this application. We consider a multiuser Wi-Fi communication system, and we use a ray tracing algorithm (WISE software [3]) to design a realistic indoor wireless system in an office located at Bell Labs Crawford Hill. We place 12 receivers and 6 transmitters across the office, in the positions respectively marked with circles and crosses in Figure 5 (all transmitters and receivers are placed at a height of 2 metres). Transmitted symbols belong to a quadrature phase-shift keying (QPSK) constellation, $\mathcal{A} = \{\frac{\pm 1 \pm \sqrt{-1}}{\sqrt{2}}\}$, such that, while active, the transmitted symbols are independent and uniformly distributed in $\mathcal{A}$, i.e., $p(x_{tm} | s_{tm} = 1, x_{(t-1)m}, s_{(t-1)m}) = \mathcal{U}(\mathcal{A})$.

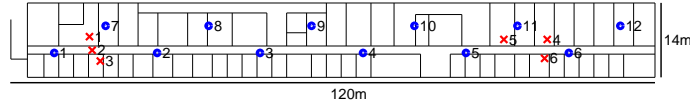

Figure 5: Plane of the office building at Bell Labs Crawford Hill.

| Model | L | | | | |
|---|---|---|---|---|---|
| | 1 | 2 | 3 | 4 | 5 |
| iFDM | 6/6 | 6/6 | 6/6 | 6/6 | 6/6 |
| iFHMM | 3/11 | 3/11 | 3/8 | 1/10 | – |

(a) # Recovered transmitters / Inferred $M_+$.

| Model | L | | | | |
|---|---|---|---|---|---|
| | 1 | 2 | 3 | 4 | 5 |
| iFDM | 2.58 | 2.51 | 0.80 | 0.30 | 0.16 |
| iFHMM | 2.79 | 1.38 | 5.53 | 1.90 | – |

(b) MSE of the channel coefficients ($\times 10^{-6}$).

Table 3: Results for the multiuser detection problem.

The observations of all the receivers are weighted replicas of the transmitted symbols under noise, $\mathbf{y}_t = \sum_{m=1}^{M_+} \sum_{\ell=1}^{L} \mathbf{h}_\ell^m x_{(t-\ell+1)m} + \mathbf{n}_t$, where $x_{tm} = 0$ for the inactive states, and the channel coefficients $\mathbf{h}_\ell^m$ and noise variance $\sigma_y^2$ are provided by WISE software. For inference, we assume Rayleigh-fading channels and, therefore, we place a circularly symmetric complex Gaussian prior distribution over the channel coefficients, $\mathbf{h}_\ell^m | \sigma_\ell^2 \sim \mathcal{CN}(\mathbf{0}, \sigma_\ell^2 \mathbf{I}, \mathbf{0})$, and over the noise term, $\mathbf{n}_t \sim \mathcal{CN}(\mathbf{0}, \sigma_y^2 \mathbf{I}, \mathbf{0})$. We place an inverse gamma prior over $\sigma_\ell^2$ with mean and standard deviation $0.01 e^{-0.5(\ell-1)}$. The choice of this particular prior is based on the assumption that the channel coefficients $\mathbf{h}_\ell^m$ are *a priori* expected to decay with the memory index $\ell$, since the radio signal suffers more attenuation as it propagates through the walls or bounces off them. We use an observation period $T = 2,000$, and vary $L$ from $1$ to $5$. Five channel taps correspond to the radio signal travelling a distance of 750 m, which should be enough given the dimensions of this office space. We compare our iFDM with a non-binary iFHMM model with state space cardinality $|\mathcal{X}| = 5^L$ using FFBS sweeps for inference (we do not run the FFBS algorithm for $L = 5$ due to its computational complexity).

We show in Table 3a the number of recovered transmitters (i.e., the number of transmitters for which we recover all the transmitted symbols with no error) found after running the inference algorithms, together with the inferred value of $M_+$. We see that the iFHMM tends to overestimate the number of transmitters, which deteriorates the overall symbol estimates and, as a consequence, not all the transmitted symbols are recovered. We additionally report in Table 3b the MSE of the first channel tap, i.e., $\frac{1}{6 \times 12} \sum_m ||\mathbf{h}_1^m - \widehat{\mathbf{h}}_1^m||^2$, being $\widehat{\mathbf{h}}_\ell^m$ the inferred channel coefficients. We sort the transmitters so that the MSE is minimized, and ignore the extra inferred transmitters. In general, the iFDM outperforms the iFHMM approach, as discussed above. Under our iFDM, the MSE decreases as we consider a larger value of $L$, since the model better fits the actual radio propagation model.

## 5   Conclusions

We have proposed a general BNP approach to solve source separation problems in which the number of sources is unknown. Our model builds on the mIBP to consider a potentially unbounded number of hidden Markov chains that evolve independently according to some dynamics, in which the state space can be either discrete or continuous. For posterior inference, we have developed an algorithm based on PGAS that solves the intractable complexity that the FFBS presents in many scenarios, enabling the application of our iFDM in problems such as multitarget tracking or multiuser detection. In addition, we have shown empirically that our PGAS approach outperforms the FFBS-based algorithm (in terms of accuracy) in the cocktail party and power disaggregation problems, since the FFBS gets more easily trapped in local modes of the posterior in which several Markov chains correspond to a single hidden source.

**Acknowledgments**
I. Valera is currently supported by the Humboldt research fellowship for postdoctoral researchers program and acknowledges the support of Plan Regional-Programas I+D of Comunidad de Madrid (AGES-CM S2010/BMD-2422). F. J. R. Ruiz is supported by an FPU fellowship from the Spanish Ministry of Education (AP2010-5333). This work is also partially supported by Ministerio de Economía of Spain (projects COMPREHENSION, id. TEC2012-38883-C02-01, and ALCIT, id. TEC2012-38800-C03-01), by Comunidad de Madrid (project CASI-CAM-CM, id. S2013/ICE-2845), by the Office of Naval Research (ONR N00014-11-1-0651), and by the European Union 7th Framework Programme through the Marie Curie Initial Training Network 'Machine Learning for Personalized Medicine' (MLPM2012, Grant No. 316861).

## Footnotes

[1]Code for these applications can be found at `https://github.com/franrruiz/iFDM`

[2]http://spandh.dcs.shef.ac.uk/projects/chime/PCC/datasets.html

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
