[Supplementary Material]

# Supplementary Material: Infinite Factorial Dynamical Model

**Isabel Valera**[*]
Max Planck Institute for
Software Systems
ivalera@mpi-sws.org

**Francisco J. R. Ruiz**[*]
Department of Computer Science
Columbia University
f.ruiz@columbia.edu

**Lennart Svensson**
Department of Signals and Systems
Chalmers University of Technology
lennart.svensson@chalmers.se

**Fernando Perez-Cruz**
Universidad Carlos III de Madrid, and
Bell Labs, Alcatel-Lucent
fernandop@ieee.org

## 1 Details on the Inference Algorithm

In **Step 1**, we first sample an auxiliary slice variable $\vartheta$, which is distributed as

$$\vartheta|\mathbf{S}, \{a^m\} \sim \text{Uniform}\left(0, a_{\min}\right), \tag{1}$$

where $a_{\min} = \min_{m:\exists t, s_{tm} \neq 0} a^m$, and we can replace the uniform distribution with a more flexible scaled beta distribution. Then, starting from $m = M_+ + 1$, new variables $a^m$ are iteratively sampled from

$$p(a^m|a^{m-1}) \propto \exp\left(\alpha \sum_{t=1}^{T} \frac{1}{t}(1-a^m)^t\right)(a^m)^{\alpha-1}(1-a^m)^T \mathbb{I}(0 \leq a^m \leq a^{m-1}), \tag{2}$$

with $a^{M_+} = a_{\min}$, until the resulting value is lesser than the slice variable, i.e., until $a^m < \vartheta$. Since Eq. 2 is log-concave in $\log a^m$ [15], we can apply adaptive rejection sampling (ARS) [6] in this step. Let $M_{\text{new}}$ be the number of new variables $a^m$ that are greater than the slice variable. If $M_{\text{new}} > 0$, then we expand the representation of matrices $\mathbf{S}$ and $\mathbf{X}$ by adding $M_{\text{new}}$ zero columns, and we sample the corresponding per-chain global variables from the prior, which depends on the specific application at hand.

In **Step 3**, we sample the global variables in the model from their complete conditional distributions.[1] The complete conditional distribution over the transition probabilities $a^m$ under the semi-ordered stick-breaking construction [15] is given by

$$p(a^m|\mathbf{S}) = \text{Beta}\left(n_{01}^m, 1 + n_{00}^m\right), \tag{3}$$

being $n_{ij}^m$ the number of transitions from state $i$ to state $j$ in the $m$-th column of $\mathbf{S}$. For the transition probabilities from active to active $b^m$, we have

$$p(b^m|\mathbf{S}) = \text{Beta}\left(\beta_0 + n_{11}^m, \beta_1 + n_{10}^m\right). \tag{4}$$

Finally, we sample the rest of global variables, including the emission or observation variables $\boldsymbol{\theta}$, from their complete conditionals, which depend on each specific application.

## 2 Experiments Description

Here we provide some details on the four considered applications, regarding the problem description, data preprocessing, hyperparameter choice, and some technical details about the inference procedure.

In all our applications, we use $P = 3,000$ particles for the PGAS kernel, and we set the mIBP hyperparameters, $\mathbf{S} \sim \text{mIBP}(\alpha, \beta_0, \beta_1)$, as $\alpha = 1$, $\beta_0 = 2$ and $\beta_1 = 0.1$. The choice of $\beta_0$ and $\beta_1$ is based on the fact that we expect the active Markov chains to remain active and, therefore, the self-transition probabilities of the active states $b^m$, which are $\text{Beta}(\beta_0, \beta_1)$ distributed, are *a priori* expected to be large.

### 2.1 Multitarget Tracking

**Brief problem description.** In the multitarget tracking problem, we aim at locating the position of several moving targets based on noisy observations. Under a general setup, a varying number of indistinguishable targets are moving around in a region, with targets arising at random in space and time. Multitarget tracking plays an important role in many areas of engineering such as surveillance, computer vision and signal processing [14, 16, 11]. Here, we focus on a simple synthetic example to show that our proposed iFDM can handle time-dependent continuous-valued hidden states. In this example, we assume that the targets are constantly transmitting radio signals that reach the sensors, although our model can handle other scenarios as well. We also consider that targets can switch on and off (i.e., start or stop transmitting) at any given time and, when they switch on, they can be at any position within the region.

**Data acquisition and preprocessing.** We generate synthetic data for this application. We place three different moving targets within a region of $800 \times 800$ metres. Sensors are located on a square grid, being 200 metres the minimum distance between two sensors. We generate $T = 300$ observations. Each transmitter becomes active at a time instant uniformly sampled in the interval $[1, T/2]$ and becomes inactive $T/2$ time instants afterwards (this ensures that the different transmitted signals overlap).

In order to generate data, we consider that the state $\boldsymbol{x}_{tm} = [x_{tm}^{(1)}, x_{tm}^{(2)}, v_{tm}^{(1)}, v_{tm}^{(2)}]^\top$ of each target consists of its position and velocity in a two dimensional plane, and we assume a linear Gaussian dynamic model, i.e., while active, $\boldsymbol{x}_{tm}$ evolves according to

$$\boldsymbol{x}_{tm} = \mathbf{G}_x \boldsymbol{x}_{(t-1)m} + \mathbf{G}_u \mathbf{u}_t = \begin{bmatrix} 1 & 0 & T_s & 0 \\ 0 & 1 & 0 & T_s \\ 0 & 0 & 1 & 0 \\ 0 & 0 & 0 & 1 \end{bmatrix} \boldsymbol{x}_{(t-1)m} + \begin{bmatrix} \frac{T_s^2}{2} & 0 \\ 0 & \frac{T_s^2}{2} \\ T_s & 0 \\ 0 & T_s \end{bmatrix} \mathbf{u}_t, \qquad (5)$$

where $T_s = 0.5$ is the sampling period and $\mathbf{u}_t$ is a vector that models the acceleration noise. We assume $\mathbf{u}_t \sim \mathcal{N}(\mathbf{0}, \mathbf{I})$, and when a transition from inactive to active occurs, the position is uniformly distributed in the sensor network space, while the velocity is Gaussian distributed with zero mean and covariance $0.01\mathbf{I}$.

Similarly to [16, 11], sensors measure the received signal strength (RSS). In other words, the observation of sensor $j$ at time $t$ is given by

$$y_{tj} = \sum_{m:s_{tm}=1} P_0 \cdot \left( \frac{d_0}{d_{mjt}} \right)^\gamma + n_{tj}, \qquad (6)$$

where $n_{tj} \sim \mathcal{N}(0, 2)$ is the noise term, $P_0 = 10$ is the transmitted power, $d_{mjt}$ is the distance between target $m$ and sensor $j$ at time $t$, and $d_0 = 100$ metres and $\gamma = 2$ are, respectively, the reference distance and the path loss exponent, which account for the radio propagation model.

**Model description.** We consider the same dynamic and likelihood models that we use to generate data. In particular, for the dynamical model, we have that[2]

$$p(\boldsymbol{x}_{tm}|s_{tm} = 1, \boldsymbol{x}_{(t-1)m}, s_{(t-1)m} = 1) = \mathcal{N}(\boldsymbol{x}_{tm}|\mathbf{G}_x \boldsymbol{x}_{(t-1)m}, \mathbf{G}_u \mathbf{G}_u^\top), \qquad (7)$$

while $p(\boldsymbol{x}_{tm}|s_{tm} = 1, \boldsymbol{x}_{(t-1)m}, s_{(t-1)m} = 0)$ is the product of a uniform distribution for the position components and $\mathcal{N}(\mathbf{0}, 0.01\mathbf{I})$ for the velocity components.

Regarding the likelihood of each 25-length observation vector $\mathbf{y}_t$, we use

$$p(\mathbf{y}_t|\mathbf{S}, \mathbf{X}, \sigma_y^2) = \prod_{j=1}^{25} p(y_{tj}|\mathbf{S}, \mathbf{X}, \sigma_y^2) = \prod_{j=1}^{25} \mathcal{N}\left(y_{tj} \left| \sum_{m:s_{tm}=1} P_0\left(\frac{d_0}{d_{mjt}}\right)^\gamma, \sigma_y^2\mathbf{I}\right.\right), \quad (8)$$

where the noise variance $\sigma_y^2$ is a global variable, over which we place an InvGamma$(1, 1)$ prior.

We compare the performance of our iFDM with a genie-aided factorial model that has perfect knowledge of the number of targets and the noise variance, using PGAS for inference.

**Inference and results.** We run $10,000$ iterations of the sampling algorithm described in Section 3 of the main paper. The inferred position of each transmitter is obtained by averaging its position for the last $2,000$ samples. In the main paper, we show the true and the inferred trajectories, as well as the temporal evolution of the position error (in absolute value). We also show that the average position error obtained by our iFDM is similar to the genie-aided approach.

## 2.2 Cocktail Party

**Brief problem description.** The cocktail party problem is a blind speech separation task, in which multiple people are speaking simultaneously, and we are interested in distinguishing the individual speech signals given a set of measurements of the mixed signals. Speakers are not continuously speaking, but they may start speaking or become silent at any given time [17].

**Data acquisition and preprocessing.** Similarly to [17], we collect data from several speakers from the PASCAL 'CHiME' Speech Separation and Recognition Challenge website.[3] The voice signal for each speaker consists of 4 sentences, which we append with random pauses in between each sentence. We artificially mix the data 10 times (corresponding to 10 microphones) with mixing weights sampled from Uniform$(0, 1)$, such that each microphone receives a linear combination of all the considered speech signals, corrupted by Gaussian noise with standard deviation $0.3$. We consider two scenarios, with 5 and 15 speakers, and subsample the data so that we learn from $T = 1,354$ and $T = 1,087$ datapoints, respectively.

**Model description.** Given the sub-sampling procedure, and following [17], we can ignore the dependencies of consecutive values of the voice signal. Hence, we assume that the latent states are distributed as
$$p(x_{tm}|s_{tm} = 1, x_{(t-1)m}, s_{(t-1)m}) = \mathcal{N}(x_{tm}|0, 2), \quad (9)$$
and $p(x_{tm}|s_{tm} = 0, x_{(t-1)m}, s_{(t-1)m}) = \delta_0(x_{tm})$, where $\delta_0(\cdot)$ denotes a point mass located at 0.

The likelihood of the observation $\mathbf{y}_t$ is given by a Gaussian distribution of the form

$$p(\mathbf{y}_t|\mathbf{S}, \mathbf{X}, \{\mathbf{w}_m\}, \sigma_y^2) = \mathcal{N}\left(\mathbf{y}_t \left| \sum_{m=1}^{M_+} \mathbf{w}_m x_{tm}, \sigma_y^2\mathbf{I}\right.\right). \quad (10)$$

where $\mathbf{w}_m \sim \mathcal{N}(\mathbf{0}, \mathbf{I})$ is the 10-dimensional weighting vector associated to the $m$-th speaker, and $\sigma_y^2 \sim \text{InvGamma}(1, 1)$ is the noise variance.

For comparisons, we apply the ICA iFHMM in [17] using FFBS sweeps for inference, with (i) $p(x_{tm}|s_{tm} = 1) = \mathcal{N}(0, 2)$ (FFBS-G), and (ii) $p(x_{tm}|s_{tm} = 1) = \text{Laplace}(0, 2)$ (FFBS-L).

**Inference and results.** The continuous variables $\mathbf{X}$ can be integrated out to improve the mixing of the inference algorithm, as detailed in [17]. We follow a similar approach, in which we integrate $\mathbf{X}$ before sampling matrix $\mathbf{S}$, and instantiate $\mathbf{X}$ by sampling from its posterior afterwards. The PGAS algorithm allows us to marginalize $\mathbf{X}$ from the likelihood in (10) for each particle to obtain $p(\mathbf{y}_t|\mathbf{S}, \{\mathbf{w}_m\}, \sigma_y^2)$, and hence particles are binary vectors in this case. In contrast, the FFBS algorithm samples each column of $\mathbf{S}$ conditioned on the current values of the remaining ones, which also requires conditioning on $\mathbf{X}$ for all but the considered column. Hence, under this approach, we integrate out one column of $\mathbf{X}$, run the FFBS over that column of $\mathbf{S}$, and resample the column of

**X** from its posterior afterwards. Since we condition on the values of the remaining columns, we can use the Laplace prior over $x_{tm}$ instead of the Gaussian prior and still follow the same procedure [17].

We run $10,000$ iterations of the inference algorithms. In order to evaluate the performance of each approach, we report in the main paper the estimated number of speakers as well as the activity detection error rate (ADER), which is computed as the probability of detecting activity (inactivity) of a speaker while that speaker is actually inactive (active). We average the estimated number of speakers and the ADER for the last $2,000$ iterations of the sampler.

## 2.3   Power Disaggregation

**Brief problem description.** The power disaggregation problem consists in, given the aggregate whole-home power consumption signal, estimating both the number of active devices in the house and the power draw of each individual device. Accurate estimation of the specific device-level power consumption avoids instrumenting every individual device with monitoring equipment, and the obtained information can be used to significantly improve the power efficiency of consumers [4, 13]. Furthermore, it allows providing recommendations about their relative efficiency (e.g., a household that consumes more power in heating than the average might need better isolation) and detecting faulty equipment.

Recently, this problem has been addressed in [9] by applying a factorial hidden semi-Markov model (HSMM) and using an expectation maximization (EM) algorithm, and in [8] using an explicit-duration hierarchical Dirichlet process HSMM. However, in both works, the number of devices in the house is assumed to be known. Furthermore, the former uses training data to learn the device models, and the latter includes prior knowledge to model each specific device.

**Data acquisition and preprocessing.** We consider two different real databases for the power disaggregation problem:

- The REDD database [10], which monitors several homes at low and high frequency for large periods of time. We consider a 24-hour segment across 5 houses and choose the low-frequency power consumption of 6 devices: refrigerator (R), lighting (L), dishwasher (D), microwave (M), washer-dryer (W) and furnace (F). We apply a 30-second median filter and scale the data dividing by 100.
- The AMP database [12], which records the power consumption of a single house using 21 sub-meters for an entire year (from April 1st, 2012 to March 31st, 2013) at one minute read intervals. We consider two 24-hours segments and choose 8 devices: basement plugs and lights (BME), clothes dryer (CDE), clothes washer (DWE), kitchen fridge (FGE), heat pump (HPE), home office (OFE), entertainment-TV, PVR, AMP (TVE) and wall oven (WOE). We scale the data by a factor of $1/100$.

**Model description.** Our model for this application is a non-binary iFHMM, in which we assume that each device can take $Q = 4$ different states (one inactive state and three active states with different power consumption), i.e., $x_{tm} \in \{0, 1, \ldots, Q-1\}$. We place a symmetric Dirichlet prior over the transition probability vectors of the form $\mathbf{a}_j^m \sim \text{Dirichlet}(1)$, being $\mathbf{a}_j^m$ the transition probability vector from state $j$ of device $m$. Hence, the dynamical model can be written as

$$p(x_{tm} = k | s_{tm} = 1, x_{(t-1)m} = j, s_{(t-1)m}, \mathbf{a}_j^m) = a_{jk}^m, \tag{11}$$

being $a_{jk}^m$ the elements of $\mathbf{a}_j^m$, and

$$p(x_{tm} | s_{tm} = 0, x_{(t-1)m}, s_{(t-1)m}) = \delta_0(x_{tm}). \tag{12}$$

The state $x_{tm} = 0$ corresponds to the inactive state and, therefore, when $x_{tm} = 0$ the power consumption of device $m$ at time $t$ is zero ($P_0^m = 0$). For the active states, when $x_{tm} \in \{1, \ldots, Q-1\}$, its average power consumption is given by $P_{x_{tm}}^m$. The total power consumption $y_t$ is assumed Gaussian distributed as

$$p(y_t | \mathbf{S}, \mathbf{X}, \{P_q^m\}) = \mathcal{N}\left(y_t \left| \sum_{m=1}^{M_+} P_{x_{tm}}^m, 0.5 \right.\right). \tag{13}$$

M (12%)
L (42%)
D (22%)
R (24%)

M (2%)
L (37%)
D (35%)
R (27%)

D (31%)
L (61%)
R (8%)

(a) Ground truth.     (b) PGAS.     (c) FFBS.

Figure 1: REDD - House 1. Percentage of the total power consumption consumed by each device.

BME (13%)   TVE (7%)   OFE (4%)
CDE (12%)
DWE (4%)
FGE (5%)
HPE (55%)

BME (19%)
CDE (19%)
FGE (6%)
HPE (56%)

BME (5%)
CDE (5%)
DWE (4%)
FGE (12%)
HPE (74%)

(a) Ground truth.     (b) PGAS.     (c) FFBS.

Figure 2: AMP - Day 1. Percentage of the total power consumption consumed by each device.

For $q \in \{1, \ldots, Q-1\}$, we assume $P_q^m \sim \mathcal{N}(15, 10)$.

**Inference and results.** We run $10,000$ iterations of our inference algorithm. Since the model resembles a non-binary iFHMM in this case, we also apply a FFBS-based algorithm for comparisons.

In order to evaluate the performance of the different algorithms, we compute the mean accuracy of the estimated consumption of each device (higher is better), which is measured as [10]

$$\text{acc} = 1 - \frac{\sum_{t=1}^{T} \sum_{m=1}^{M} |x_t^{(m)} - \hat{x}_t^{(m)}|}{2 \sum_{t=1}^{T} \sum_{m=1}^{M} x_t^{(m)}}, \tag{14}$$

where $x_t^{(m)}$ and $\hat{x}_t^{(m)} = P_{x_{tm}}^m$ are, respectively, the true and the estimated power consumption by device $m$ at time $t$. In order to compute the accuracy, we assign each estimated chain to a device so that the accuracy is maximized. If the inferred number of devices $M_+$ is smaller than the true number of devices, we use $\hat{x}_t^{(m)} = 0$ for the undetected devices. If $M_+$ is larger than the true number of devices, we group all the extra chains as an "unknown" device and use $x_t^{(\text{unk})} = 0$.

The obtained accuracy values, averaged for the last $2,000$ iterations of the sampler, are reported in the main paper. Here, we additionally show the true percentage of total power consumed by each device, together with the inferred percentages (also averaged for the last $2,000$ iterations), for both inference algorithms. In particular, Figures 1 and 2 show the results for House 1 of the REDD database and Day 1 of the AMP database, respectively. In these plots, we have considered that any percentage below $1\%$ is negligible.

## 2.4 Multiuser Detection

**Brief problem description.** When digital symbols are transmitted over communication channels, inter-symbol interference (ISI) may occur, degrading the performance of the receiver. To improve the performance, channel estimation is applied to mitigate the effects of ISI. Blind channel estimation involves channel estimation (typically jointly with symbol detection) without the use of training data. We address the problem of blind joint channel parameter and data estimation in a multiuser communication channel in which the number of transmitters is not known. In the literature, we can find several works addressing this problem [19, 7, 20, 3, 1, 2, 18]. However, a characteristic shared by all of them is the assumption of an explicit upper bound for the number of transmitters (users), which may represent a limitation in some scenarios. Our BNP approach naturally avoids this limitation by assuming instead an unbounded number of transmitters.

**Data acquisition and preprocessing.** We focus on a Wi-Fi multiuser communication channel. Wi-Fi systems are not limited by the noise level, which is typically negligible, but by the user interferences, which can be avoided by using a particular frequency channel for each user. Our goal is to show that cooperation of receivers in a Wi-Fi communication system can help recover the symbols transmitted by several users even when they simultaneously transmit over the same frequency channel, therefore allowing for a larger number of users in the system.

We use WISE software [5] to design a realistic indoor wireless system in an office located at Bell Labs Crawford Hill. WISE software, developed at AT&T Bell Laboratories, includes a 3D ray-tracing propagation model, as well as algorithms for computational geometry and optimization, to calculate measures of radio-signal performance in user-specified regions. Its predictions have been validated with physical measurements [5].

The bandwidth of the Wi-Fi system is 20 MHz or, equivalently, 50 ns per channel tap. We place 12 receivers and 6 transmitters across the office, intentionally placing the transmitters together in order to ensure that interferences occur in the nearby receivers. We simulate the transmission of $1,000$-symbol bursts over this communication system, using a QPSK constellation normalized to yield unit energy. We scale the channel coefficients by a factor of 100, and we consequently multiply the noise variance by $10^4$, yielding $\sigma_y^2 \approx 7.96 \times 10^{-9}$. We set the transmission power to 0 dBm. Each transmitter becomes active at a random point, uniformly sampled in the interval $[1, T/2]$, and we consider an observation period of $T = 2,000$.

The observations of all the receivers are generated as $\mathbf{y}_t = \sum_{m=1}^{M_+} \sum_{\ell=1}^{L} \mathbf{h}_\ell^m x_{(t-\ell+1)m} + \mathbf{n}_t$, being $\mathbf{h}_\ell^m$ for $\ell = 1, \ldots, L$ the channel coefficients as provided by WISE software, $x_{tm}$ the transmitted symbols (or zero if transmitter $m$ is inactive at time instant $t$), and $\mathbf{n}_t \sim \mathcal{CN}(\mathbf{0}, \sigma_y^2\mathbf{I}, \mathbf{0})$ the additive noise term.[4]

**Model description.** While active, the transmitted symbols are independent and uniformly distributed in the QPSK constellation $\mathcal{A} = \{\frac{\pm 1 \pm \sqrt{-1}}{\sqrt{2}}\}$, so we set

$$p(x_{tm}|s_{tm} = 1, x_{(t-1)m}, s_{(t-1)m}) = \mathcal{U}(\mathcal{A}), \tag{15}$$

where $\mathcal{U}(\mathcal{A})$ stands for the uniform distribution over the set $\mathcal{A}$. For the inactive states, we use $p(x_{tm}|s_{tm} = 0, x_{(t-1)m}, s_{(t-1)m}) = \delta_0(x_{tm})$.

The observations of all the receivers are weighted replicas of the transmitted symbols under complex Gaussian noise, i.e.,

$$p(\mathbf{y}_t|\mathbf{S}, \mathbf{X}, \{\mathbf{h}_\ell^m\}) = \mathcal{CN}\left(\mathbf{y}_t \left| \sum_{m=1}^{M_+} \sum_{\ell=1}^{L} \mathbf{h}_\ell^m x_{(t-\ell+1)m}, \sigma_y^2\mathbf{I}, \mathbf{0}\right.\right). \tag{16}$$

We assume Rayleigh-fading channels and, therefore, we place a circularly symmetric complex Gaussian prior distribution over the channel coefficients, $\mathbf{h}_\ell^m|\sigma_\ell^2 \sim \mathcal{CN}(\mathbf{0}, \sigma_\ell^2\mathbf{I}, \mathbf{0})$. We place an inverse

gamma prior over $\sigma_\ell^2$ with mean and standard deviation $0.01e^{-0.5(\ell-1)}$. The choice of this particular prior is based on the assumption that the channel coefficients $\mathbf{h}_m^\ell$ are *a priori* expected to decay with the memory index $\ell$, since the radio signal suffers more attenuation as it propagates through the walls and bounces off them.

**Inference and results.** In our experiments, we vary $L$ from 1 to 5. Five channel taps correspond to the radio signal travelling a distance of 750 m, which should be enough given the dimensions of this office space (the signal suffers attenuation when it reflects on the walls, so we should expect it to be negligible in comparison to the line-of-sight ray after a 750-m travelling distance). We compare our iFDM with a non-binary iFHMM model with state space cardinality $|\mathcal{X}| = 5^L$ using FFBS sweeps for inference (we do not run the FFBS algorithm for $L = 5$ due to its computational complexity).

We observed in our experiments a poor performance in terms of error rates. However, we found that we could improve the performance by considering a communication channel with higher noise variance or, equivalently, a lower signal-to-noise ratio (SNR). This counter-intuitive effect can be easily understood by taking into account the posterior distribution and the inference procedure. When the SNR is high enough, the noise variance is too small compared to the variance of the channel coefficients, which makes the posterior get narrow around the true value of these coefficients. In other words, the posterior uncertainty on the channel coefficients becomes small, and similarly for the transmitted symbols. As a consequence, an inference algorithm based on random exploration of the posterior needs more iterations to find the peaks of the posterior distribution. In practice, we cannot afford such large number of iterations. Instead, we propose a solution based on an heuristic to artificially widen the posterior distribution. For that purpose, we add artificial noise to the observations, consequently decreasing the SNR. From an "exploration versus exploitation" perspective, this method eases exploration of the posterior. At each iteration of the algorithm, we slightly increase the SNR by reducing the variance of the artificial noise, and we repeat this procedure until we reach the actual value of the noise variance. After that, we run additional iterations to favor exploitation. In our experiments, we linearly increase the SNR for around $26,600$ iterations, running $3,400$ additional iterations with fixed SNR afterwards.

As a metric, we measure the MSE of the first channel tap, i.e., $\frac{1}{6\times 12}\sum_m ||\mathbf{h}_1^m - \widehat{\mathbf{h}}_1^m||^2$, averaged for the last $2,000$ iterations of the algorithm, and report the obtained results in the main paper. We also show in the main paper the number of inferred transmitters $M_+$, as well as the number of recovered transmitters (also averaged for the last $2,000$ iterations), where we say that a transmitter has been recovered if the symbol error rate is equal to zero. In order to match the inferred transmitters to the true ones, we sort the transmitters so that the MSE is minimized.

## Footnotes

[1]The complete conditional is the conditional distribution of a hidden variable, given the observations and the rest of hidden variables.

[2]Note that the covariance matrix specified in Eq. 7 is singular, but we keep this notation for simplicity.

[3]http://spandh.dcs.shef.ac.uk/projects/chime/PCC/datasets.html

[4]The complex Gaussian distribution over a vector $\mathbf{x}$ of length $D$, denoted as $\mathcal{CN}(\boldsymbol{\mu}, \boldsymbol{\Gamma}, \mathbf{C})$, is given by $p(\mathbf{x}) = \frac{1}{\pi^D \sqrt{\det(\boldsymbol{\Gamma})\det(\mathbf{P})}} \exp\left\{-\frac{1}{2}\left[(\mathbf{x}-\boldsymbol{\mu})^{\mathsf{H}}, (\mathbf{x}-\boldsymbol{\mu})^{\top}\right] \begin{bmatrix} \boldsymbol{\Gamma} & \mathbf{C} \\ \mathbf{C}^{\mathsf{H}} & \boldsymbol{\Gamma}^\star \end{bmatrix}^{-1} \begin{bmatrix} \mathbf{x}-\boldsymbol{\mu} \\ (\mathbf{x}-\boldsymbol{\mu})^\star \end{bmatrix}\right\}$, where $\mathbf{P} = \boldsymbol{\Gamma}^\star - \mathbf{C}^{\mathsf{H}}\boldsymbol{\Gamma}^{-1}\mathbf{C}$, $(\cdot)^\star$ denotes the complex conjugate, and $(\cdot)^{\mathsf{H}}$ denotes the conjugate transpose. A circularly symmetric complex Gaussian distribution has $\boldsymbol{\mu} = \mathbf{0}$ and $\mathbf{C} = \mathbf{0}$.