[Reviews · NeurIPS 2015]

Submitted by Assigned_Reviewer_1

The authors introduce a new Bayesian nonparametric model, more specifically an infinite factorial dynamic model (iFDM). This model is based on the Markov Indian buffet process and it allows for a possibly unbounded number of hidden Markov chains, each of which evolve independently according to some dynamics. The resulting inference problem is solved using particle Gibbs with ancestor sampling. The application in mind is that of source separation and the approach is evaluated on four examples; multitarget tracking, cocktail party, power disaggregation, and multiuser detection. The iFDM contains several interesting simpler models as special cases.

Why is the state of the underlying Markov chain restricted to be of order one? There is a comment in the multitarget tracking example that the existing BNP approaches cannot handle this. What is the problem exactly?

You are using a lot of particles, do you experience problems with degeneracy?

You can probably improve the performance quite a lot by making use of a fully adapted filter, rather than the bootstrap as you are currently doing. Since your models are analytically tractable I think this should be quite straightforward to derive. More specifically this boils down to changing equation (4) and possibly also the resampling step. An early and basic introduction is provided in (Section II.D)

http://www.stats.ox.ac.uk/~doucet/doucet_godsill_andrieu_sequentialmontecarloforbayesfiltering.pdf where it is referred to as the optimal proposal. If you google ``fully adapted particle filters'' you will find a lot more material.

The authors have considered four different and all relevant application examples. The experimental section shows that the iFDM seems to work and that it can provide interesting results. The only comparison provided is against the FFBS-type algorithm, which we know will perform worse due to its construction. I know that it is a lot of work to implement other solutions to the problem, but if one were to do so it would probably provide an even better understanding of the performance of the model and it would be interesting to see the performance of existing solution to these problems. For example, for the multitarget tracking example, the simplest solution to this problem would probably be to use an extended Kalman filter together with nearest neighbour data association. Since your targets are very well separated I would expect this solution to perform quite well. It would be interesting to compare your performance against this simple standard solution. I have not worked with the cocktail party problem and the multiuser detection problems, but for the power disaggregation problem there are interesting solutions available, see for example the following NIPS paper (which is gaining some influence):

Kolter, J. Z.; Batra, S.; and Ng, A. Y. Energy disaggregation via discriminative sparse coding. In Advances in Neural Information Processing Systems (NIPS). 2010. and the subsequent variational inference solution Kolter, J. Z., and Jaakkola, T. Approximate inference in additive factorial hmms with application to energy disaggregation. International Conference on Artificial Intelligence and Statistics. 2012.
Summary: This is a well written and clear paper showing how to make use of a recent MCMC kernel for inference in an interesting new

Bayesian nonparametric model. As an application example the authors make use of various source separation problem.

Submitted by Assigned_Reviewer_2

The paper proposes a flexible nonparametric sequence modeling framework entitled the "infinite factorial dynamical model" (iFDM). iFDM generalizes existing infinite factorial models for discrete hidden spaces, such as the iFHMM, and models with strong independence assumptions, such as iICA. It also extends the class of nonparametric hidden space models to allow continuous spaces, which could not be modeled previously, or high dimensionality discrete hidden spaces, for which efficient inference was not possible. Efficient inference have been developed based on parallel Gibbs with ancestor sampling (PGAS), an elaboration of particle filtering.

The paper presents results from very thorough experiments on four different applications that require very different models: LDS for multi-target tracking; two blind source separation problems using ICA and non-binary HMMs, respectively; and high dimensional binary state HMMs for multi-user tracking. In each case, the iFDM beats a strong baseline (often a special case of the iFDM) in terms of both modeling accuracy and estimation of the number of targets/sources/etc.

The paper is pleasant to read. The work is rigorous and thorough. It should be of interest to the NIPS community. The experimental results are quite convincing. It would be useful to address how to make the model scale to really large datasets in the final version.
Summary: This paper describes a new nonparametric sequence model that generalizes existing models, extends the class of infinite hidden state models to new problems, and admits a tractable inference algorithm based on parallel Gibbs with ancestor sampling (PGAS). Experimental results show that this flexible model can be applied to a broad set of problems and often beats existing approaches, even those that are special cases of the proposed framework.

The proposed solution is not significantly novel given that there is a large literature in related areas. However, I appreciate that its efforts to unify a broad class of sequence models and conduct extensive experiments on a variety of application datasets.

Submitted by Assigned_Reviewer_3

This paper proposes a hidden Markov model in which an infinite number of hidden chains evolve independently and at each step a subset of the combine to produce the output (which depends on the last k steps of the relevant Markov chains).

A Markov Indian buffet process is used to model the "relevant subset" at each time point.

Partical Gibbs with ancestor sampling is used as the key resampling step inside the model.

Four experiments are provided in a variety of related but distinct domains. This represents good coverage.

Most experiments are synthetic in some respect.

The power disaggregation task is done on real data.

The cocktail party example uses real data, but the mixing is done synthetically. The other two are generated completely synthetically.

All experiments are compared to different versions of the same model and algorithms.

No comparisons are given to domain-specific solutions.

In all four experiments, there are existing solutions.

To judge the utility of this new formulation, we should see how the proposed method compares to existing solutions.

Eq 1 and proceeding remark might be better stated as a set of (conditional) independences?

Paper is well written.

Thank you.
Summary: This is an interesting combination of existing ideas.

The experimental results are good in their scope (four different domains), but lack comparison to existing methods for these domains.

Submitted by Assigned_Reviewer_4

The paper presents a generalisation of the infinite factorial HMM (Van Gael et al. 2009) which can handle (1) non-binary (i.e., arbitrary discrete or continuous) state spaces, and (2) system memory in the emission function. These extensions are necessary to address many problems in signal processing and related fields. Thus, even though it is a fairly straightforward extension of the iFHMM, the proposed model has the potential to be very useful to the community. Inference in the new model is enabled via the recent Particle Gibbs with ancestor sampling (PGAS) MCMC algorithm.

* The model is evaluated on four numerical examples. The presentation of the experiments is quite brief, and it is hard to assess the actual performance of the new model. You might want to consider moving one or two of the numerical examples completely to the supplemental material in order to more clearly describe and show the empirical results for the the ones that are included in the main paper.

* If I understand correctly, when you sample the (X,S) matrices using the PGAS algorithm, you treat the M parallel chains as a joint system state. This might be problematic if M is large, in particular when using the bootstrap proposal density. One possibility is to update "a few" (e.g., selected randomly) of the chains and update only these using PGAS, with the other chains kept fixed. Have you considered this option?

* You use a fixed number of P=3000 particles in the numerical examples. Did you try with a smaller P? Lindsten et al. (2014) argue in their paper that PGAS can be used with a "small" number of particles.

* In the multitarget tracking example you say that "existing BNP models cannot handle the dynamical model". However, many models and inference techniques have been proposed in the literature for the MTT problem, which can handle continuous states, unknown number of targets, target births and deaths, etc. You should acknowledge this body of work and compare against at least one state-of-the-art MTT method (see e.g., Jiang et al. (2014) or Saerkkae et al. (2007)).

* The proposed inference algorithm is far from trivial to implement, and I strongly encourage the authors to provide software to make the new model useful to the community.

References:

Lan Jiang, Sumeetpal S. Singh, Sinan Yildirim, "Bayesian tracking and parameter learning for non-linear multiple target tracking models", arXiv:1410.2046, 2014.

Simo Saerkkae, Aki Vehtari1, Jouko Lampinen, "Rao-Blackwellized particle filter for multiple target tracking", Information Fusion 8(1):2-15, 2007.
Summary: The paper presents a generalisation of the infinite factorial HMM, resulting in a quite general Bayesian nonparametric dynamical model (allowing for continuous latent states and system memory). The generalisation is fairly straightforward, but it significantly broadens the scope of the model and therefore it has the potential to widely used. This is illustrated by a numerical evaluation on several diverse examples.

Author Feedback
Author rebuttal: We sincerely thank the reviewers for their comments on the paper, which will help us to make it stronger and to continue extending this work.

All reviewers agree that this is an interesting and well-written paper. A general concern raised by several reviewers is the lack of comparisons with application-specific approaches, in addition to our current comparisons with FFBS-based methods. Besides the space constraints, we decided not to compare with such approaches for fairness: in general, state-of-the-art methods cannot consider an unbounded number of sources, so we restricted our attention to methods with similar abilities as our iFDM. However, we agree that comparisons with application-specific approaches would strengthen the paper, making it more complete. We will consider that for the revised version of the paper.

We reply below to other specific questions and comments raised by the reviewers.

Reviewer_1:
We consider a first order Markov chain because it simplifies the BNP construction of the model and the inference, while being flexible enough to capture the dynamics of the four considered applications. The model could be easily extended to account for a higher order Markov chain, although it would make inference computationally more demanding.

Regarding the multitarget tracking problem, we would like to remark that the existing BNP approaches can only capture the temporal dependencies among the hidden binary states s_t (which indicate when a source is active or not), but not among the states x_t, which are assumed to be independent on x_{t-1}. This constitute a limitation in some application such as multitarget tracking, in which the current position of a target depends on its previous position. We will try to clarify this in the final version of the paper.

We have not observed a degeneracy problem of the particles in any of our four applications. We used P=3000 particles because that number provides a good tradeoff for all our applications, with either discrete or continuous hidden states.

Thank you for your comment and references. It is true that a fully adapted filter would allow reducing the number of particles while keeping good mixing properties, but it is not clear whether we can actually generate particles for all the models that we consider. We will study this in future extensions of the paper.

Reviewer_2:
We would like to remark that Eq. 1 simply states that y_t is conditionally independent of the hidden states at all other times, given the hidden states from time t-L+1 up to t. We believe the representation Eq. 1 is readable, although there may be other ways to convey the same idea, and therefore, we will consider other options for the revised version of the paper.

Reviewer_3:
We agree the reviewer that when M is large, we would need a large number of particles in order for the PGAS algorithm (in which we sample all the chains jointly) to mix well, and in that case a straightforward solution is to sequentially sample from subsets of chains. In other words, instead of increasing the number of particles, we can improve mixing by updating only a few chains at a time. We have considered that, and we have even followed this approach in other experiments, not included in the paper.

Regarding the complexity of the algorithm, evaluating (6) for each tau is O(PML). However, as tau ranges from t up to t+L-2, the resulting computational cost is O(PML^2).

We used P=3000 particles because that number provides a good tradeoff for all our applications, although we would like to point out that smaller values of P also work in some scenarios.

We will upload the code to our website. Thank you for your recommendation.

Reviewer_4:
We agree with the reviewer that it would be very useful to derive more scalable inference algorithms to handle very large datasets, and this is an idea we have in mind for future future extensions of this work.